# Impact of Magnetic Biostimulation and Environmental Conditions on the Agronomic Quality and Bioactive Composition of INIA 601 Purple Maize

**DOI:** 10.3390/foods14122045

**Published:** 2025-06-10

**Authors:** Tony Chuquizuta, Cesar Lobato, Franz Zirena Vilca, Nils Leander Huamán-Castilla, Wilson Castro, Marta Castro-Giraldez, Pedro J. Fito, Segundo G. Chavez, Hubert Arteaga

**Affiliations:** 1Instituto de Investigación del Mejoramiento Productivo, Universidad Nacional Autónoma de Chota, Chota 06120, Peru; tchuquizuta@unach.edu.pe (T.C.); 2016041020@unach.edu.pe (C.L.); 2Instituto Universitario de Ingeniería de Alimentos Food UPV, Universitat Politècnica de València, Camino de Vera s/n, 46022 Valencia, Spain; marcasgi@upv.es (M.C.-G.); pedfisu@tal.upv.es (P.J.F.); 3Laboratory of Organic Pollutants and Environment of the IINDEP of the Universidad Nacional de Moquegua, Urb Ciudad Jardín-Pacocha-Ilo, Moquegua 18001, Peru; fzirenav@unam.edu.pe; 4Laboratorio de Tecnologías Sustentables para la Extracción de Compuestos de Alto Valor, Instituto de Investigación para el Desarrollo del Perú, Universidad Nacional de Moquegua, Moquegua 18001, Peru; nhuamanc@unam.edu.pe; 5Facultad de Ingeniería de Industrias Alimentarias y Biotecnología, Universidad Nacional de Frontera, Sullana 20100, Peru; wcastro@unf.edu.pe; 6Instituto de Investigación para el Desarrollo Sustentable de Ceja de Selva INDES-CES, Universidad Nacional Toribio Rodríguez de Mendoza de Amazonas, Chachapoyas 01001, Peru; 7Grupo de Investigación Innovación Tecnológica en Productos y Procesos Alimentarios (GIITPA), Instituto de Investigación de Ciencia y Tecnología de Alimentos (ICTA), Universidad Nacional de Jaén, Jaén 06800, Peru

**Keywords:** antioxidants, Andean crops, biofortification strategies, bioactive yields in plants, phenolic biosynthesis

## Abstract

The utilization of magnetic fields in agricultural contexts has been demonstrated to exert a beneficial effect on various aspects of crop development, including germination, growth, and yield. The present study investigates the impact of magnetic biostimulation on seeds of purple maize (*Zea mays* L.), variety INIA 601, cultivated in Cajamarca, Peru, with a particular focus on their physical characteristics, yield, bioactive compounds, and antioxidant activity. The results demonstrated that seeds treated with pulsed (8 mT at 30 Hz for 30 min) and static (50 mT for 30 min) magnetic fields exhibited significantly longer cobs (16.89 and 16.53 cm, respectively) compared with the untreated control (15.79 cm). Furthermore, the application of these magnetic fields resulted in enhanced antioxidant activity in the bract, although the untreated samples exhibited higher values (110.56 µg/mL) compared with the pulsed (91.82 µg/mL) and static (89.61 µg/mL) treatments. The geographical origin of the samples had a significant effect on the physical development and the amount of total phenols, especially the antioxidant activity in the coronet and bract. Furthermore, a total of fourteen phenols were identified in various parts of the purple maize, with procyanidin B2 found in high concentrations in the bract and crown. Conversely, epicatechin, kaempferol, vanillin, and resveratrol were found in lower concentrations. These findings underscore the phenolic diversity of INIA 601 purple maize and its potential application in the food and pharmaceutical industries, suggesting that magnetic biostimulation could be an effective tool to improve the nutritional and antioxidant properties of crops.

## 1. Introduction

The significance of purple corn extends both nationally and internationally. During the initial quarter of 2024, the marketing of grain, bracts, and crowns enabled the export of 281 tons of corn on the cob. The United States constitutes its primary market [1]. This variety of corn is utilized in the production of various baked goods, gastronomy, whiskey, and in the extraction of anthocyanin compounds. Noteworthy among the most cultivated varieties are the following: Cuzco, Caraz, Arequipa, Negro de Junín, Huancavelicano, UNC-47, PM-581, PM-582, INIA-615 black Canaan, and INIA-601. The latter variety is of particular interest due to its bioactive compounds, particularly anthocyanin, which possesses significant antioxidant activity. This bioactive compound has been shown to offer numerous health benefits, including a reduction in the risk of developing diabetes [2]. It has been demonstrated that it is instrumental in preventing obesity and being overweight, colon cancer and cardiovascular diseases, and ocular degeneration and contributes to the formation of cells and tissues. The efficacy of these benefits has been substantiated by numerous research studies. Conversely, improper seed selection has been shown to result in diminished levels of phytochemicals, such as anthocyanins and flavonoids, which are pivotal for the health benefits associated with purple corn [3]. This, in turn, has been observed to diminish the quality and economic value of purple corn [4].

In the pursuit of enhancing the germination and physical characteristics of corn, various technologies have been employed, including precision agriculture, which utilizes specialized software and hardware such as GPS and Geographic Information Systems (GISs). These technologies enable farmers to meticulously manage inputs, including fertilizers, pesticides, and irrigation water, thereby minimizing waste and reducing the environmental impact [5]. Conversely, Genetically Modified Organisms (GMOs) have been linked to an augmentation in the area and yield of maize [6]. Another method is ultrasonic priming of seeds, in which ultrasonic waves at about 20 kHz are applied in water or other suitable nutrient solutions to improve germination and early growth [7] as well as uniformity [8]. However, due to its competitive cost advantages, its varied application, and its positive effects on crops, the application of magnetic fields stands out. These fields have been shown to improve growth and yield, mitigate abiotic stress factors that affect vegetative development [9] and seed quality [10], and increase the availability of some minerals, such as phosphorus, in the soil [11].

Magnetic fields have been demonstrated to promote the germination and growth of diverse agricultural seeds. The interaction of magnetic fields with biological systems within the spectrum below 1 MHz is referred to as charge induction, which involves the transfer of charges from ions at low frequencies to the induction of metal centers (Fe^3+^, Cu^2+^, and Mn^2+^) in proteins that play a crucial role in cellular functioning at higher frequencies. Molecules such as ferritins, plastocyanins, cytochrome C oxidase, and superoxide dismutase, which are crucial for plant growth [12], are affected by paramagnetic relaxation in seeds. In the cases of maize and soybean, an intensity of 200 mT for 1 h has been shown to enhance germination and essential enzyme activities, even in saline soils [13]. Furthermore, the application of 10 mT at 25 °C for 1 h to brown rice seeds resulted in an augmentation of α-amylase activity and germination, along with enhancements of shoot and root length and weight [14]. In broad beans, this treatment increased amylolytic activity and production of indole-3-acetic and gibberellic acids [15]. Furthermore, water subjected to magnetic fields exhibited a propensity to promote maize growth, yielding outcomes that surpassed those of conventional water [16]. The underlying mechanism of this phenomenon is attributed to the ability of magnetic fields to modify apical cells, thereby enhancing nutrient uptake [17]. However, these treatments have not yet been applied to purple corn seeds of the INIA 601 variety. Consequently, this study evaluated the effect of magnetic biostimulation of INIA 601 purple corn seeds on their physical characteristics, production yield, concentration of bioactive compounds, and antioxidant activity.

## 2. Materials and Methods

### 2.1. Vegetal Material

The experiment utilized 3 kg of purple corn (*Zea mays* L. var. INIA 601) seeds, kindly provided by the “Baños del Inca” Experimental Station of the Instituto Nacional de Innovación Agraria (INIA), located in Cajamarca, Peru.

### 2.2. Regents

Hydrochloric acid, ethanol (99.7%), potassium chloride (0.025 M), sodium acetate, concentrated hydrochloric acid, Folin Ciocalteu, gallic acid, sodium carbonate (95.5%), formic acid, and methanol (99.7%) were obtained from Merck KGaA (Darmstadt, Alemania); 2,2-diphenyl-1-picrylhydrazyl (DPPH) was also acquired (Burlington, MA, USA), along with HPLC-grade methanol (JT Baker, Madrid, Spain).

### 2.3. Experimental Location

The experiment was conducted at two locations in the department of Cajamarca. These were Cajabamba and Cochamarca, whose geographical locations and climatological characteristics for 2022 and 2023 growing seasons are shown in Table 1.

### 2.4. Methods

#### 2.4.1. Biostimulation of INIA 601 Purple Corn Seeds

The static and pulsed magnetic field prototypes were designed and constructed at the Emerging Technologies Laboratory of the Research Institute of the Universidad Nacional Autónoma de Chota and the Food Engineering Laboratory of the Universidad Nacional de Jaén. The static magnetic field (SMF) prototype consisted of an array of neodymium magnets, which were placed on PLA printed plates. The pulsed magnetic field (PMF) prototype consisted of a solenoid coil, pulse controllers, and a voltage source. Prior to the application of the magnetic fields, the seeds were meticulously selected, ensuring the exclusion of any seeds exhibiting signs of mechanical or biological damage. The seeds were subsequently transferred to the equipment for biostimulation with static magnetic fields, with an induction of 50 mT for 30 min and pulsed magnetic fields at a frequency of 30 Hz for 30 min (Figure 1).

In both experimental plots, a control treatment was installed that was not subjected to a magnetic field (NMF). All experimental plots were prepared and conducted in a homogeneous manner so that the only sources of variation were the treatments and the location.

#### 2.4.2. Evaluation of the Physical Characteristics of INIA 601 Purple Corn

From the six central furrows of each experimental unit, the days to 50% male flowering, the days to 50% female flowering, and the number of ears harvested per plant were recorded. From ten randomly selected plants, the plant height and ear height were determined; from ten randomly selected ears from the six central furrows, the ear length was measured. A six-class scale devised by the Maize and Wheat Research Center-CIMMYT [18]. was used to ascertain the percentage of rotting.

#### 2.4.3. Production Yield of INIA 601 Purple Corn

The production yield of INIA 601 purple corn was determined in accordance with the procedure established by Medina et al. [19]. This procedure involved the consideration of various factors, including 14% grain moisture, the field weight, the failure correction factor, the failure factor, the shelling factor, and the factor used to express the yield in tons per hectare (t/ha).

#### 2.4.4. Obtaining INIA 601 Purple Corn Extracts

The method of Lao and Giusti [20] was used, with some modifications. Cobs were shelled to separate bracts, crowns, and kernels; they were reduced in size in a multifunctional pulverizer (Henkel, WFA-GR1000, Düsseldorf, Germany) and passed through a N° 35 sieve (500 µm). The moisture of the samples was determined using a moisture balance (AND MX-50) at 105 °C; 1 g of each sample was used, along with 20 mL of 50% ethanol solvent acidified with 0.01% hydrochloric acid (HCl) at 6 N, placed in a magnetic stirrer (Cimarec+ Thermo Scientific, Saint Louis, MO, USA) at 350 rpm for 30 min at 30 °C, and centrifuged (CENTURION-Scientific Limited, Chichester, UK) at 4000 rpm for 10 min. The supernatant was filtered using Wattman No. 1 paper, placed in amber containers, and stored at −24 °C.

#### 2.4.5. Determination of Total Anthocyanin Content (TAC)

The total anthocyanin content was determined via the differential pH method described by Lee et al. [21], using potassium chloride buffer 0.025 M at pH 1 and sodium acetate 0.4 M at pH 4.5. The extracts obtained were as follows: for the corn cob, the dilution factor (DF) was 100, with 30 µL of extract used; for the bract, the DF was 50, with 60 µL of extract used; for the grain, the DF was 10, with 300 µL of extract needed. The difference for each extract was augmented with buffers up to 3 mL. The samples were vortexed for 1 min and left to react for 30 min in the absence of light at room temperature. Subsequently, the absorbance was read at 520 and 700 nm using a Genesys 150 UV–Visible spectrophotometer. It is imperative to note that each sample was subjected to analysis in triplicate.

#### 2.4.6. Determination of Total Phenol Content (TPC)

The Folin–Ciocalteu method, as described by Albarici et al. [22] and Jin et al. [23]), was employed to ascertain the total phenol content. The experimental setup involved the utilization of 0.2 mL of the diluted samples, 0.1 mL of Folin–Ciocalteu reagent, and 0.2 mL of 10% sodium carbonate. These components were allowed to react in a dark environment for a duration of 1 h. Subsequently, the reaction was measured at 765 nm using a Genesys 150 UV–Visible spectrophotometer. Prior to this, a standard curve was constructed using different dilutions of gallic acid to obtain a linear equation (y: a + bx; R^2^ > 0.998). The results were expressed as mg of AGE/g db.

#### 2.4.7. Profile of Phenolic Compounds by HPLC

The experimental procedure was executed in accordance with the protocol established by Ramos-Escudero et al. [24]. The extracts were subjected to analysis using an ultra-high-performance chromatograph (UHPLC) (Agilent 1290 Infinity II, Santa Clara, CA, USA) equipped with a Diode Array Detector (DAD) and a Poroshell C18 column. The analysis was conducted at 30 °C. The chromatographic separation was achieved through the use of a mobile phase consisting of acetonitrile and 0.1% formic acid (phase A) and water with 0.1% formic acid (phase B). The separation was conducted using a programmed gradient, which involved an initial 95% A + 5% B for 15 min, followed by a transition to 60% A + 40% B for 18 min. The final step involved a return to 95% A + 5% B for 20 min, with all steps occurring at a constant flow rate of 0.3 mL/min. The results were expressed as µg/g db. Table 2 shows the detection and quantification limits for the phenolic compounds studied. The calibration curves had an R^2^ greater than 0.995.

#### 2.4.8. Determination of Antioxidant Activity

The quantification of the 2,2 diphenyl, 1 picrylhydrazyl (DPPH) radical uptake method was performed in accordance with the methodology established by Ratha et al. [25]), with certain modifications. The extraction process involved the dilution of 1.0 mL of the extract with methanol (200 µM), followed by the addition of 1 mL of the methanolic solution of DPPH. Subsequently, the mixture was permitted to stand for a period of 30 min in conditions of darkness. Ultimately, readings were obtained at 517 nm utilizing a Genesys 150 UV–Visible spectrophotometer. The IC50 was determined from the graph of the percentage of inhibition versus the concentrations used to inhibit 50% of the free radicals of DPPH. The results were expressed as µg sample/mL of the DPPH solution.

#### 2.4.9. Statistical Analysis

The data were tabulated and presented in figures prepared in Microsof Excel spreadsheets. To determine statistical differences between groups (treatments), an analysis of variance (ANOVA) and Tukey’s post hoc test were performed, using Statgraphics Centurion V.19 software.

## 3. Results

### 3.1. Physical Characteristics

The application of magnetic fields to purple corn seeds did not yield substantial modifications in the male flowering day (MFD) and female flowering day (FFD), plant height, number of corn cobs per plant, degree of rotting, and yield. However, significant variations in corn-cob length were observed, with treatments subjected to magnetic fields exhibiting lengths exceeding 16.5 cm. The planting locations exerted a substantial influence on the characteristics previously mentioned, with the exception of corn-cob length. The Cajabamba experimental plot exhibited the most optimal outcomes in terms of MFD, FFD, and plant height (Table 3), attributable to its higher sunlight exposure, which averaged 5.4 h per day (Table 1). In contrast, the Cochamarca plot demonstrated superior characteristics in the number of corn cobs per plant (prolificacy), the incidence of root lodging and stem lodging, and yield. This was due to the fact that taller plants were obtained in Cajabamba, which were more prone to the effects of weather conditions, resulting in a high percentage of root and stem lodging (20.8 and 10%, respectively).

### 3.2. Bioactive Composition

As demonstrated in Figure 2, the magnetic fields exerted an influence on the total phenol content of the bracts of INIA 601 purple corn. The most effective treatment was identified as CMP, yielding 28.253 mg EAG/gss. The planting locations exhibited a significant impact on the phenol content, with Cajabamba and Cochamarca registering levels of 25 and 28 mg EAG/g db, respectively.

As illustrated in Figure 3, the application of magnetic fields to INIA 601 purple corn seeds did not result in substantial variations in total anthocyanin content. Given the greater variation in temperatures between day and night at Cochamarca (see Table 1), there was a propensity for increased anthocyanin content in the bract and corn cob, with levels of 7.6 mg and 9.4 mg of Cyanidin-3-O-glucoside (C3G)/gss, respectively. This was in comparison to the samples from Cajabamba, which exhibited 6.5 mg of C3G/gss in the bract and 9.3 mg of C3G/gss in the cob.

### 3.3. Phenolic Compound Profile

In Figure 4, we can observe the phenols found in the INIA 601 purple corn ear kernels. The most important were procyanidin B2, which exceeded 25 mg/L, and resveratrol, which exceeded 15 mg/L. In addition, the treatment with the pulsed magnetic field was able to increase the amount of catechin compared with the other treatments. The mean values and standard deviations are presented in Table 4.

Figure 5 shows the phenols present in the crown of INIA 601 purple corn, among which procyanidin B2 stood out, which exceeded 40 mg/L, as well as resveratrol, aspecinine, and catechin, the latter being affected by the pulsed magnetic field of square waves.

Figure 6 shows that the phenols present in the INIA 601 purple corn bract. A considerable amount of procyanidin B2 was found, exceeding 50 mg/L in the samples that had CMP applied, as well as resveratrol, which exceeded 10 mg/L. In addition, two phenols, epicatechin and tyrosol, were found only in the bract compared with the grain and coronet samples.

### 3.4. Antioxidant Activity

As illustrated in Figure 7, the magnetic fields exerted a substantial influence on the bract, with CME and CMP requiring 89.612 µg and 91.816 µg/mL of DPPH solution, respectively, in comparison to SCM, which necessitated an extract concentration 110.556 µg/mL higher to achieve the equivalent inhibition of free radicals. In the cob, significant differences were also observed, with CMP requiring 161.136 µg/mL compared with CME and SCM. In the grain, no significant differences were observed; however, a slight variation was noted, with CMP yielding a more favorable result.

The geographical provenance of the samples exhibited a substantial influence on the outcomes, with the bract samples from Cochamarca demonstrating superior antioxidant activity, with an average of 81.5 µg/mL, in comparison to those from Cajabamba, which exhibited an average of 113.081 µg/mL.

## 4. Discussion

The outcomes for the corn-cob length (16.5 cm) aligned with those reported by Huaychani [26]. The length of a corn cob is closely associated with the optimal utilization of fertilizers containing nitrogen, phosphorus, and potassium as well as heightened sunlight intensity linked to elevated photosynthesis stimulation [27]. The impact of the insect *Euxesta* sp. is particularly problematic as it attacks the ears from the pistil formation stage and continues during the development of the grain [19]. This resulted in a high rotting percentage of 19.99% in Cajabamba compared with Cochamarca, which was 9.65%. The aforementioned factors directly affected the low yield (3.35 t/ha) in Cajabamba compared with Cochamarca (4.89 t/ha). However, high fertilization is possible, with yields of up to 6.6 t/ha according to the results reported by Huaychani (2022) [26].

The observed variation in phenol content could be attributed to the temperature fluctuations experienced during the day and night, which have been shown to induce stress in the crop, thereby stimulating the biosynthesis of anthocyanin and phenolic compounds in the crown and the pod of the cob [27]. Previous studies have examined the impact of magnetic fields on soybean, revealing their regulatory role in protein metabolism and leading to an augmentation in globulin and lipid production [28]. Additionally, these fields have been observed to enhance the levels of phenols and flavonoids in the shoots of lemon balm [29] and concentrate the mineral compounds in certain parts of a plant, as observed in Castlerock tomato, where P was found in the leaves of the plant [30].

Given the high content of bioactive compounds found in purple corn on the cob, it is necessary to specify that foods high in phenolic compounds are responsible for the stimulation of muscle tissue and the production of nitric oxide, necessary for the dilation of blood vessels [31].

Ramos-Escudero et al. [24] obtained an extract of INIA 601 purple corn grain with 80% methanol and 20% water and acidified with 1% HCl, and identified eight phenolic compounds. These were chlorogenic acid, caffeic acid, rutin, ferulic acid, morin, quercetin, naringenin, and kaempferol, the latter being the highest at 2240 mg/kg of the sample. Such studies highlight the presence of procyanidin B2 as the predominant phenol in various parts of INIA 601 purple corn, although the extraction and analytical methods used could influence the detection of other specific phenolic compounds.

The antioxidant activity is associated with the variety of plant samples utilized, the solvents employed, and the extraction techniques used for the bioactive compounds [32]. The outcomes significantly diverged from those reported by Ramos-Escudero et al. [24], who conducted an extraction using 80% methanol and 20% water acidified with 1% HCl (1 N), yielding a DPPH concentration of 66.3 µg/mL for the INIA 601 variety.

Variations by place of culture can be attributed to the diurnal temperature fluctuations, which have been observed to enhance bioactive compounds and, consequently, antioxidant activity [27].

Recent studies suggest that phytohormones have a direct influence on the gene expression of the biosynthetic pathways related to flavonoids and anthocyanins [33,34]. According to Shan et al. [35] the JAZ protein (jasmonate) is ubiquitinated by COI1 and subsequently degraded by the 26S proteasome, thereby promoting the release of bHLH and MYB transcription factors and reforming the stable MBW complex. This complex, in turn, promotes anthocyanin synthesis and accumulation in plants. Similarly, the auxin concentration has been observed to exert a promoting or inhibiting effect on anthocyanin production, with this effect being contingent upon the specific plant species [36]. A comparable phenomenon has been documented in the context of the phytohormone gibberellin [37]. Subsequent research endeavors could involve the investigation of the impact of magnetic fields on the activity of these phytohormones.

Finally, these findings allow the demonstration of the future applications of magnetic biostimulation in the high production of bioactive compounds in purple corn cultivation under different agro–climatic conditions. A larger experiment with a larger number of experimental units and different conditions could better clarify the results found in this work.

## 5. Conclusions

The utilization of magnetic fields on INIA 601 purple corn seeds prior to planting did not exert a substantial influence on the agronomic characteristics under consideration, with the exception of the length of the corn cob, which exceeded 16.5 cm. Furthermore, it was associated with the adequate fertilization of the soil. Conversely, the planting location exerted an influence on the crop characteristics, with Cajabamba exhibiting a greater propensity to experience adverse conditions, thereby exerting a negative effect on yield, in contrast to Cochamarca, which demonstrated superior results due to more favorable environmental conditions and a reduced incidence of pests and diseases.

The application of magnetic fields demonstrated the exertion of a substantial influence on the concentration of bioactive compounds, including total phenols and anthocyanins, within the corn cob. Conversely, the planting location exerted a discernible influence on the phenol content, with Cajabamba exhibiting 25.6 mg AGE/gss and Cochamarca registering 28.6 mg AGE/gss. Temperature variations were observed to stimulate anthocyanin biosynthesis, with higher levels recorded in the bract in Cochamarca (7.6 mg C3G/gss) as well as in the crown (9.4 mg C3G/gss). In contrast, lower levels of anthocyanin were detected in Cajabamba, with 6.5 mg C3G/gss in the bract and 9.3 mg C3G/gss in the crown. These findings underscore the pivotal role of environmental and climatic factors on the synthesis and composition of bioactive compounds in INIA 601 purple corn.

The analysis of 12 phenol compounds in various parts of the cob of INIA 601 purple corn highlighted the presence of procyanidin B2, especially in the bract and the crown, where it exceeded 50 and 40 mg/L, respectively. The analysis also identified the presence of resveratrol, epicatechin, kaempferol, and vanillin, among other phenols, although in lower concentrations. These findings underscore the remarkable diversity of phenols in INIA 601 purple corn across its various corn cobs, which could have significant ramifications for its utilization in the food and pharmaceutical sectors.

## Figures and Tables

**Figure 1 foods-14-02045-f001:**
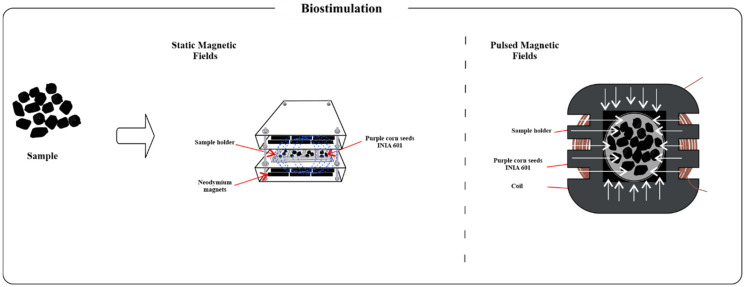
Application of static and pulsed magnetic fields to INIA 601 purple corn seeds.

**Figure 2 foods-14-02045-f002:**
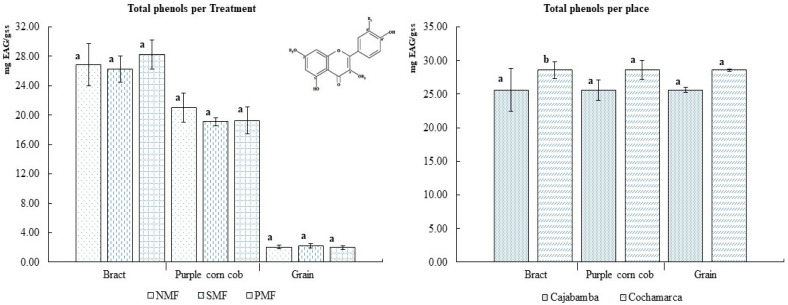
Total phenol content of INIA 601 purple corn samples. NMF: nonmagnetic field; SMF: static magnetic field; PMF: pulse magnetic field. Different letters indicate statistically different groups (*p* ≤ 0.05).

**Figure 3 foods-14-02045-f003:**
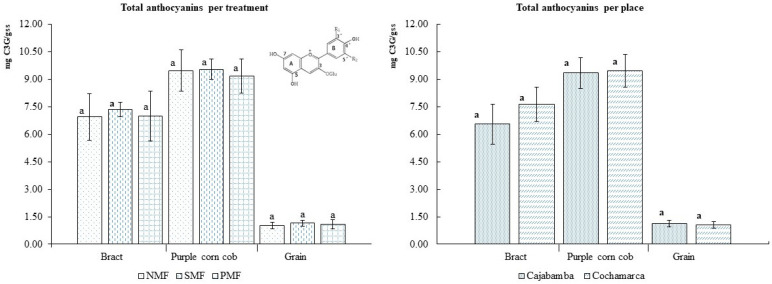
Total anthocyanin content of INIA 601 purple corn samples. NMF: nonmagnetic field; SMF: static magnetic field; PMF: pulse magnetic field. Different letters indicate statistically different groups (*p* ≤ 0.05).

**Figure 4 foods-14-02045-f004:**
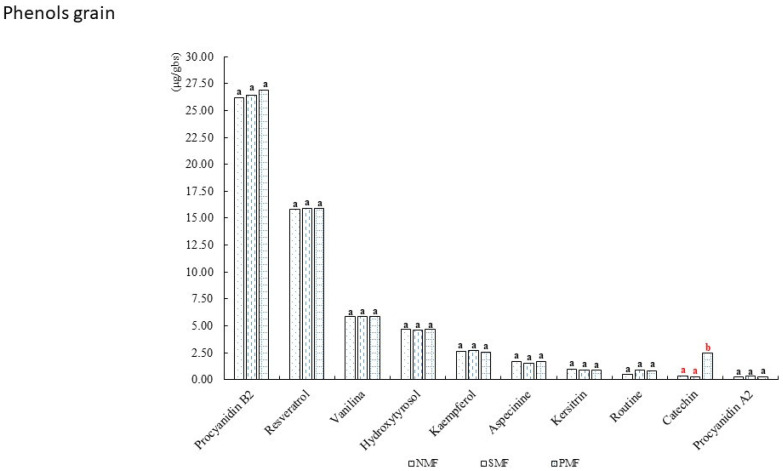
Phenolic content of INIA 601 purple corn kernels. NMF: nonmagnetic field; SMF: static magnetic field; PMF: pulse magnetic field. Different letters indicate statistically different groups (*p* ≤ 0.05).

**Figure 5 foods-14-02045-f005:**
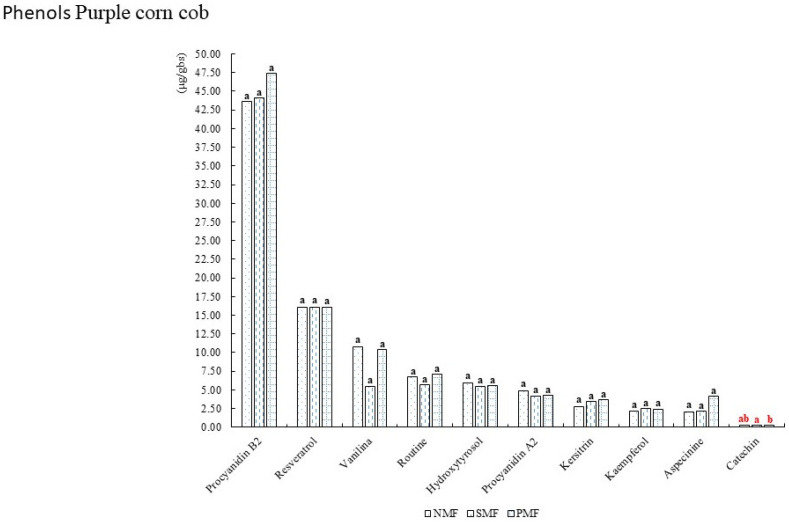
Phenols of INIA 601 purple corn crowns. NMF: nonmagnetic field; SMF: static magnetic field; PMF: pulse magnetic field. Different letters indicate statistically different groups (*p* ≤ 0.05).

**Figure 6 foods-14-02045-f006:**
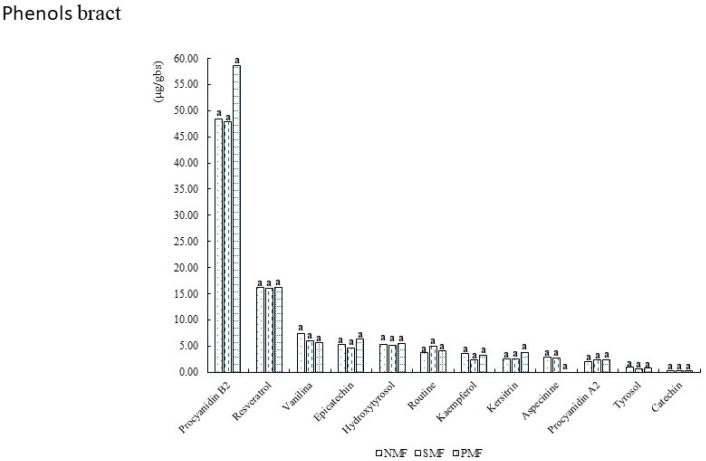
Phenols of INIA 601 purple corn bract. NMF: nonmagnetic field; SMF: static magnetic field; PMF: pulse magnetic field. Different letters indicate statistically different groups (*p* ≤ 0.05).

**Figure 7 foods-14-02045-f007:**
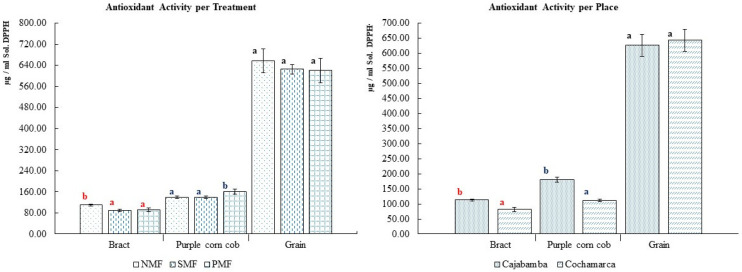
Antioxidant activity of INIA 601 purple corn samples. NMF: nonmagnetic field; SMF: static magnetic field; PMF: pulse magnetic field. Different letters and color indicate statistically different groups (*p* ≤ 0.05).

**Table 1 foods-14-02045-t001:** Geographical location and climatological characteristics of the planting locations.

Location	Political Location		Geographic Coordinates			
			Altitude (masl)	Latitude		Longitude
Cajabamba	Plot of land of INIA Pampa Grande Annex, district and province of Cajabamba		2640	7°36′39″		78°04′15″
Cochamarca	INIA land lot, Gregorio Pita district, San Marcos province		2864	7°16′52″		78°13′09″
	**Climatic characteristics**					
	**Minimum daily temperature** **(°C)**	**Maximum daily temperature** **(°C)**	**Average daily precipitation (mm)**	**Daily relative humidity** **(%)**	**Hours of sun/day** **(h)**	**Wind speed** **(m/s)**
Cajabamba	12.82	23.33	3.53	74.96	5.47	0.37
Cochamarca	7.94	20.49	2.87	82.22	3.20	0.60

**Table 2 foods-14-02045-t002:** LOD and LOQ of the HPLC system for the phenolic compounds studied.

Compound	LOD (µg mL^−1^)	LOQ (µg mL^−1^)
Hydroxytyrosol	0.0017	0.0051
Tyrosol	0.0002	0.0007
Catechin	0.0029	0.0087
Procyanidin B2	0.0057	0.0171
Epicatechin	0.0009	0.0026
Vanillin	0.0003	0.0008
Routine	0.0001	0.0003
Procyanidin A2	0.0005	0.0014
Resveratrol	0.0076	0.0229
Kersetrin	0.0088	0.0264
Aspecinine	0.0007	0.0022
Kaempferol	0.0049	0.0147

**Table 3 foods-14-02045-t003:** Physical characteristics of INIA 601 purple corn.

Variable Means				MFD (Days)	FFD (Days)	Plant Height (m)	N° of Cobs/Plant	Cob Length (cm)	Rot (%)	Root Canker (%)	Stem Canker (%)	Yield (t/ha)
**Treatment**		**NMF**	$\bar{x}$	99.100 ^a^	105.100 ^a^	2.245 ^a^	0.849 ^a^	15.791 ^a^	14.992 ^a^	7.448 ^a^	7.208 ^a^	4.155 ^a^
			**SD**	2.211	2.809	0.055	0.073	0.429	3.356	2.639	3.979	0.699
		**SMF**	$\bar{x}$	98.800 ^a^	103.400 ^a^	2.304 ^a^	0.856 ^a^	16.526 ^b^	13.982 ^a^	14.515 ^a^	8.102 ^a^	4.107 ^a^
			**SD**	1.506	2.125	0.118	0.086	0.481	3.625	7.426	2.701	0.738
		**PMF**	$\bar{x}$	98.400 ^a^	103.000 ^a^	2.290 ^a^	0.871 ^a^	16.889 ^b^	15.491 ^a^	10.383 ^a^	7.713 ^a^	4.117 ^a^
			**SD**	1.587	1.842	0.098	0.091	0.426	3.676	2.802	3.634	0.944
**Place**	**Cajabamba**		$\bar{x}$	92.200 ^a^	98.933 ^a^	2.342 ^b^	0.723 ^a^	16.475 ^a^	19.992 ^b^	20.837 ^b^	10.846 ^b^	3.353 ^a^
			**SD**	2.709	3.786	0.102	0.111	0.534	4.938	7.627	5.171	0.828
	**Cochamarca**		$\bar{x}$	105.333 ^b^	108.733 ^b^	2.217 ^a^	0.994 ^b^	16.329 ^a^	9.651 ^a^	0.727 ^a^	4.503 ^a^	4.899 ^b^
			**SD**	0.827	0.730	0.078	0.055	0.356	2.167	0.950	1.704	0.759
*p*-Value (Place)				0.000	0.000	0.000	0.000	0.393	0.000	0.000	0.000	0.000
*p*-Value (Treatment)				0.720	0.165	0.233	0.866	0.000	0.665	0.071	0.864	0.991

NMF: nonmagnetic field; SMF: static magnetic field; PMF: pulse magnetic field; MFD: male flowering day; FFD: female flowering day; $\bar{x}$: average; SD: standard deviation. Different letters indicate statistically different groups (*p* ≤ 0.05).

**Table 4 foods-14-02045-t004:** Content of 12 phenolic compounds in corn cobs produced from each treatment and cultivation site (µg/gdb).

Part of the Corn Cob	Phenolic Compound	Cajabamba			Cochamarca		
		NMF	SMF	PMF	NMF	SMF	PMF
Grain	Tyrosol	*	*	*	*	*	*
	Routine	*	*	*	0.696 ± 0.043	1.089 ± 0.264	0.938 ± 0.299
	Kersetrin	0.882 ± 0.092	0.865 ± 0.046	0.923 ± 0.01	1.038 ± 0.151	0.928 ± 0.01	0.911 ± 0.026
	Hydroxytyrosol	4.695 ± 0.039	4.585 ± 0.129	4.65 ± 0.005	4.635 ± 0.006	4.674 ± 0.027	*
	Catechin	0.29 ± 0.129	0.19 ± 0.012	*	*	*	*
	Procyanidin B2	24.954 ± 1.574	26.157 ± 0.821	26.038 ± 0.507	27.408 ± 1.414	27.683 ± 1.678	26.839 ± 1.959
	Epicatechin	*	*	*	*	*	*
	Vanilina	5.903 ± 0.171	5.921 ± 0.182	5.807 ± 0.058	5.819 ± 0.106	5.884 ± 0.087	5.974 ± 0.112
	Procyanidin A2	0.309 ± 0.153	0.277 ± 0.084	0.256 ± 0.142	0.275 ± 0.022	0.338 ± 0.051	0.231 ± 0.159
	Resveratrol	*	*	15.88 ± 0.024	15.9 ± 0.003	15.934 ± 0.053	15.978 ± 0.106
	Aspecinine	1.039 ± 0.312	1.106 ± 0.208	1.601 ± 0.499	2.274 ± 0.631	1.897 ± 0.706	1.771 ± 0.526
	Kaempferol	2.16 ± 0.334	2.679 ± 0.197	2.514 ± 0.349	3.142 ± 0.627	2.798 ± 0.412	2.637 ± 0.483
Purple corn cob	Tyrosol	*	*	*	*	*	*
	Routine	*	*	*	0.696 ± 0.043	1.089 ± 0.264	0.938 ± 0.299
	Kersetrin	0.882 ± 0.092	0.865 ± 0.046	0.923 ± 0.01	1.038 ± 0.151	0.928 ± 0.01	0.911 ± 0.026
	Hydroxytyrosol	4.695 ± 0.039	4.585 ± 0.129	4.65 ± 0.005	4.635 ± 0.006	4.674 ± 0.027	*
	Catechin	0.29 ± 0.129	0.19 ± 0.012	*	*	*	*
	Procyanidin B2	24.954 ± 1.574	26.157 ± 0.821	26.038 ± 0.507	27.408 ± 1.414	27.683 ± 1.678	26.839 ± 1.959
	Epicatechin	*	*	*	*	*	*
	Vanilina	5.903 ± 0.171	5.921 ± 0.182	5.807 ± 0.058	5.819 ± 0.106	5.884 ± 0.087	5.974 ± 0.112
	Procyanidin A2	0.309 ± 0.153	0.277 ± 0.084	0.256 ± 0.142	0.275 ± 0.022	0.338 ± 0.051	0.231 ± 0.159
	Resveratrol	*	*	15.88 ± 0.024	15.9 ± 0.003	15.934 ± 0.053	15.978 ± 0.106
	Aspecinine	1.039 ± 0.312	1.106 ± 0.208	1.601 ± 0.499	2.274 ± 0.631	1.897 ± 0.706	1.771 ± 0.526
	Kaempferol	2.16 ± 0.334	2.679 ± 0.197	2.514 ± 0.349	3.142 ± 0.627	2.798 ± 0.412	2.637 ± 0.483
Bract	Tyrosol	0.976 ± 0.651	0.613 ± 0.282	0.985 ± 0.582	0.878 ± 0.307	0.764 ± 0.168	0.511 ± 0.336
	Routine	3.697 ± 0.293	3.375 ± 1.175	6.025 ± 2.843	4.405 ± 0.994	1.343 ± 0.788	3.954 ± 0.422
	Kersetrin	3.049 ± 0.025	3.233 ± 0.839	3.905 ± 0.788	3.028 ± 0.226	2.959 ± 0.759	3.474 ± 0.237
	Hydroxytyrosol	5.145 ± 0.181	5.054 ± 0.062	5.741 ± 1.324	5.522 ± 0.751	5.311 ± 0.289	5.305 ± 0.376
	Catechin	0.336 ± 0.063	0.343 ± 0.105	0.345 ± 0.035	0.275 ± 0.034	0.338 ± 0.032	0.268 ± 0.115
	Procyanidin B2	42.133 ± 2.873	40.484 ± 1.53	37.286 ± 1.289	41.687 ± 0.352	37.685 ± 1.71	41.668 ± 1.329
	Epicatechin	*	*	6.762 ± 6.86	*	6.523 ± 6.322	5.934 ± 5.957
	Vanilina	10.114 ± 2.438	5.218 ± 0.407	5.658 ± 1.135	6.625 ± 1.775	6.806 ± 2.451	5.677 ± 0.41
	Procyanidin A2	1.857 ± 0.003	2.414 ± 0.347	2.24 ± 0.246	2.155 ± 0.502	2.292 ± 1.121	2.35 ± 0.061
	Resveratrol	16.276 ± 0.115	16.096 ± 0.047	16.17 ± 0.245	16.131 ± 0.169	16.064 ± 0.066	16.1 ± 0.027
	Aspecinine	1.869 ± 0.503	*	*	*	*	*
	Kaempferol	2.552 ± 0.704	2.268 ± 0.323	4.347 ± 3.586	4.714 ± 4.453	2.576 ± 0.5	2.103 ± 0.101

* No value. Value averages ± SD. NMF: nonmagnetic field; SMF: static magnetic field; PMF: pulse magnetic field; MFD: male flowering day; FFD: female flowering day.

## Data Availability

The original contributions presented in the study are included in the article, further inquiries can be directed to the corresponding author.

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
