# Peer review of "Impact of Magnetic Biostimulation and Environmental Conditions on the Agronomic Quality and Bioactive Composition of INIA 601 Purple Maize"

_foods, 2025, doi:10.3390/foods14122045_

Round 1
Reviewer 1 Report
Comments and Suggestions for Authors
This paper explores the effect of magnetic field treatments on purple corn. The topic is interesting and may be useful for improving crop quality, but the paper has several issues that need to be addressed before it can be published. Below are my main comments and suggestions for improvement.
- The HPLC results are missing important details. To my opinion parameters like calibration curves, R², LOD, and LOQ should be added in the form of a table. Without this information, accuracy of the phenolic compound measurements is problematic.
- I think it’s important to include a list of all the chemicals and reagents used in the experiments as a separate section
- It's hard to tell whether the changes in phenolic content and plant traits are due to the magnetic field or just the differences between the two locations. More information is needed about how the treatments were applied at each site, and whether all samples—treated and untreated—were handled the same way. This would help clear up the confusion and make the results more reliable.
- Key results—like the 14 phenolic compounds—are only mentioned. It would be much clearer to show this data in a table with concentrations and standard deviations.
- The discussion relies heavily on literature and does not adequately interpret the experimental findings in relation to the applied magnetic field treatments. The authors should focus more on explaining the observed trends and mechanisms supported by their own data.
- If 14 phenols are mentioned in the conclusion, that’s even more reason to show them clearly in a table.
Author Response
The authors are grateful for the reviewer's suggestion and consequently we have made the following changes:
1. we included in the respective subsection the respective additional information.
2. A subsection with the list of reagents used was included.
3. We made the respective clarifications and presented additional information.
4. We included table 4 with the mean values and standard deviations of phenolic compounds quantified.
5. We made the respective improvements.
6. The respective table was included.
Reviewer 2 Report
Comments and Suggestions for Authors
This paper noted Impact of Magnetic Biostimulation and Environmental Conditions on the Agronomic Quality and Bioactive Composition of Purple Maize INIA 601 is relatively reasonable and innovative. However, there are still some problems with the presentation of the article.
The detailed comments are listed as following:
- The experiment set up an untreated control group (SCM), which did not mention whether it was stored and processed under the same environmental conditions, except for not receiving magnetic field treatment.
- In terms of experimental methods, it is recommended to increase the parameters of magnetic biological stimulation, as there may be other better parameter combinations.
- It is recommended to add full name annotations for abbreviations in Table 2 to improve the readability of the article, such as DFM, DFF, SCM, CME, CMP.
- Please write the text in bold format for Table 1 correctly.
- Please adjust the all figures in the text to enhance its definition and academic appeal.
- Suggest deleting blank page 12 to increase academic value.
- Please provide the website citations for the references.
- Please standardize the citation format of references, such as APA and MLA.
- Please delete the “pdf” after reference 1.
- The specific content of Table 5 is not provided in the article. Please supplement the data.
- It is suggested to add other antioxidant indicators (such as ABTS, ORAC, etc.). The article only uses the DPPH radical scavenging experiment to assess antioxidant activity, and a single indicator may not comprehensively reflect the antioxidant capacity of purple corn.
- The article mentions the use of 50% ethanol and 0.01% hydrochloric acid for extraction, as well as a static magnetic field of 50 mT and a pulsed magnetic field of 8 mT at 30 Hz. Please explain in detail why this combination was chosen and whether it achieved optimal results.
- The article suggests exploring the impact of magnetic field treatment on the long-term growth and yield of purple corn. For example, whether continuous annual use of magnetic field treatment affects soil fertility or the genetic traits of the plants.
- The article's experimental planting locations are limited to two sites (Cajabamba and Cochamarca). The sample size is small, and the planting locations are singular, which may not fully reflect the general impact of magnetobiological stimulation on purple corn.
Author Response
- In section 2.3.1 the respective clarification is added.
- We greatly appreciate the reviewer's suggestion; it will certainly be very useful for our new projects.
- Full names of the acronyms used were included and the corresponding corrections were made.
- The correction was made in table 1.
- The correction was made.
- The correction was made.
- All references have URL or DOI of the source.
- We used MDPI style for cites and references.
- Apologies, but the full URL of the source was provided so that it can be retrieved.
- Since the information is pressented in a figure 7, the authors consider that inlcusion of a table would be unnecessary.
-
The authors agree with the reviewer's assessments; however, further analysis is not possible at this time. While a single technique may have limitations, this is complemented by the total phenolic content and phenolic profile determined by HPLC.
- The formula of the solvent used was based on the procedure used by Lao and Giusti (2018), as cited in the respective section.
- No action required.
- The authors appreciate and accept the limitations of the study; therefore, recommendations for conducting larger experiments are included in the discussion.
Reviewer 3 Report
Comments and Suggestions for Authors
The manuscript Impact of magnetic biostimulation and environmental conditions on the agronomic quality and bioactive composition of Purple Maize INIA 601’ by Chuquizuta et al. has been reviewed.
The paper describes the use of magnetic fields as possible biostimulants for purple maize seeds. To verify the suitability of the method, the authors examined both physical and chemical characteristics of maize. The number and length of cobs, roots and yield were not modified, in contrast the cob length was substantially different. Some differences have been found in antioxidant content.
The argument is of interest, the literature is adequate, but the results are not clearly reported.
Major revisions:
Experimental data: the statistical analysis is not described, please include a description reporting the software used
The abbreviations need to be clearly defined. The figures are unreadable.
To make the paper clearer for the readers the abbreviations should be reported below figures and tables.
Chemical analysis: My major concerns are related to the absence of a quantification of phytohormones (auxins, gibberellins, jasmonic acid..). In my opinion a dedicated HPLC analysis should be more interesting with respect to antioxidants.
HPLC analysis of antioxidant compounds should be reported in results and properly discussed.
Author Response
The authors greatly appreciate the reviewer's suggestions for improvement. We have carefully carried out each of his recommendations. We include a section with the statistical analysis information of the data; the full names of the abbreviations were included and the respective corrections were made, Table 3 is included with the average values and standard deviation of the content of the 14 phenolic compounds studied.

Round 2
Reviewer 1 Report
Comments and Suggestions for Authors
The revised version is significantly improved. You have appropriately addressed the previous concerns by including a detailed table of detected and quantified phenolic compounds, relevant validation parameters, and information on reagents used. With these additions, along with other rewritten and clarified sections, the manuscript is now acceptable for publication.
Author Response
The authors appreciate your valuable contribution.
Reviewer 2 Report
Comments and Suggestions for Authors
The author should responded the comments in details, for example, demostrate the details of your corrections and demostrate the which Line is corected. And the titiles of different references should be uniformed, such as the capital or lower-case letter.
Author Response

(The authors gave the same response as above.)

Reviewer 3 Report
Comments and Suggestions for Authors
I appreciate the modifications made. The major modification required was:
"Chemical analysis: My major concerns are related to the absence of a quantification of phytohormones (auxins, gibberellins, jasmonic acid..). In my opinion a dedicated HPLC analysis should be more interesting with respect to antioxidants."
I understand that this kind of analysis requires additional efforts and time. Due to the importance of this aspect, please include in the manuscript at least a discussion about it
Author Response
The authors greatly appreciate the suggestion. We have included a paragraph in the discussion section on the suggested topic